# Adoptive Immunotherapy for Prophylaxis and Treatment of Cytomegalovirus Infection

**DOI:** 10.3390/v14112370

**Published:** 2022-10-27

**Authors:** Christopher P. Ouellette

**Affiliations:** Division of Pediatric Infectious Diseases and Host Defense Program, Nationwide Children’s Hospital, Columbus, OH 43205, USA; christopher.ouellette@nationwidechildrens.org; Tel.: +614-722-4452; Fax: +614-722-4458

**Keywords:** cytomegalovirus, adoptive immunotherapy, virus-specific T-cells

## Abstract

Cytomegalovirus (CMV), a member of the Herpesviridae family, is frequent among hematopoietic cell transplant (HCT) and solid organ transplant (SOT) recipients in absence of antiviral prophylaxis, and is a major cause of morbidity and mortality in these vulnerable populations. Antivirals such ganciclovir, valganciclovir, and foscarnet are the backbone therapies, however drug toxicity and antiviral resistance may render these agents suboptimal in treatment. Newer therapies such as letermovir and maribavir have offered additional approaches for antiviral prophylaxis as well as treatment of drug resistant CMV infection, though may be limited by cost, drug intolerance, or toxicity. Adoptive immunotherapy, the transfer of viral specific T-cells (VSTs), offers a new approach in treatment of drug-resistant or refractory viral infections, with early clinical trials showing promise with respect to efficacy and safety. In this review, we will discuss some of the encouraging results and challenges of widespread adoption of VSTs in care of immunocompromised patients, with an emphasis on the clinical outcomes for treatment and prophylaxis of CMV infection among high-risk patient populations.

## 1. Introduction

Cytomegalovirus (CMV) is a ubiquitous virus responsible for significant mortality and morbidity in immunocompromised patients [1]. CMV infection affects 60–80% of solid organ transplant (SOT) and 70% of seropositive hematopoietic cell transplant (HCT) recipients in the absence of antiviral prophylaxis [2,3,4]. Risk factors associated with CMV reactivation include seropositivity of the recipient, reduced intensity conditioning, graft-versus-host disease (GVHD), human leukocyte antigens (HLA-B14, HLA-DRB1*01 and HLA-DRB1*13), donor activating killer immunoglobulin-like receptors (KIR) and use of high dose corticosteroids [2,3,5]. For SOT recipients, patients receiving lung and small intestine transplants have the highest risk of CMV disease, while heart recipients have a lower risk and renal transplant recipients have the lowest risk [6]. Among HCT and SOT recipients, a complex interplay between CMV infection, transplant type, GVHD and graft rejection, and the resultant net state of immunosuppression exists, creating a challenging environment of both infection and immunomodulatory management [7]. Current management focuses on reduction in immunosuppression and initiation of antiviral therapies including ganciclovir, cidofovir, foscarnet, and valganciclovir [8]. Use of these agents, while necessary, carries substantial side effect profiles including marrow suppression and nephrotoxicity. Furthermore, the development of resistance to antiviral drugs via mutations to CMV UL97 or UL54 genes further limits the efficacy of these agents against CMV [8]. Although the incidence of drug-resistant CMV remains low (0–8%) [9], it can be as high as 14.5% in high risk populations such as haploidentical HCT receiving antiviral prophylaxis [10]. Letermovir, recently FDA approved for antiviral prophylaxis for CMV post HCT, has not been approved for treatment of drug-resistant CMV [11]. Maribavir, a potent, selective nucleoside analog with activity among UL97 resistant CMV isolates, was recently approved by the FDA for treatment of refractory or resistant CMV infection, however, is not without issues surrounding drug intolerance or development of antiviral resistance [12,13]. While numerous antiviral therapies are available and have improved outcomes for CMV infection in immunocompromised patient populations, none address a primary issue with relation to viral disease in immunocompromised patient populations: the delay or failure of T-cell immune reconstitution. Given the high relevance to management of viral disease, promoting immune reconstitution via adoptive immunotherapy serves as a logical approach to treatment. Adoptive immunotherapy, specifically the transfer of viral-specific T-cells (VST) towards the virus of interest (ex. CMV), overcomes barriers of medical intolerance or antiviral resistance. Initial approaches using adoptive immunotherapy for treatment of infection included use donor lymphocyte infusions, however this approach (by nature of using a non-specific product) is complicated by high rates of GVHD (~60%) [14]. Advances in generation of viral specific T-cell products, though antigen selection or by ex vivo expansion, have demonstrated encouraging results with respect to efficacy and safety profiles [15,16,17,18]. In this review, we will focus on the use of adoptive immunotherapy for treatment and prophylaxis of CMV infection in immunocompromised patient populations.

## 2. Pathophysiology of CMV Infection and Rationale for Adoptive Immunotherapy

Cytomegalovirus, a betaherpesvirus and member of the *Herpesviridae*, infects a wide variety of cell types, establishing latency with episodes of periodic reactivation. Primary CMV infection induces a significant proinflammatory response (including interleukin-18 and interferon-ɣ (IFNɣ)) from innate immune cells [19]. NK cells perform a critical role in early CMV infection management, and defects of NK cell function are known to predispose to severe *Herpesviridae* infections [20]. Additionally, use of NK cells as adoptive immunotherapy concomitant to HCT has demonstrated an ancillary benefit in reduction of rates of CMV reactivation [21]. While the innate immune responses help temporize infection, adaptive immune responses, in particular CMV specific CD4+ and CD8+ T-cells, are critical in control of CMV infection [22]. Deficiencies or suboptimal responses from T-cell compartments have been observed to result in severe infections from herpesviruses [23]. Individuals with severe-combined immune deficiency are at life-threatening risk of severe, disseminated CMV infection [24]. Additionally, individuals with infection due to human immunodeficiency virus are at greatest risk with low CD4+ T-cell quantitative values, predisposing to risk of severe or fulminant disease [25]. Clinically, the use of functional cell-mediated immunity assessments in both HCT and SOT recipients can used in risk-assessment for viral reactivation [26,27,28]. In total, while the application of antiviral therapy has undoubtedly reduced the morbidity and mortality associated with CMV disease, reconstitution of immunity, particularly the adaptive immune response, is critical in long-term virologic control of CMV in those with compromised immune systems. 

## 3. The Infusional Product: Weighing the Pros and Cons

The immunobiology of VST generation has been discussed in several other review articles, highlighting the intricate yet subtle differences between VST products [29,30]. Here, we will briefly discuss the generalities of the VST product generation, including pertinent advantages and limitations of each approach. 

*Autologous or donor derived, ex vivo expanded VST*: in this approach, the subject’s (or original stem cell donor) peripheral blood mononuclear cells (PMBCs) are collected, and through repetitive antigenic (ex. pp65) and cytokine (ex. IL-2) stimulation, CMV T-cell populations are expanded ex vivo [17,31]. The resultant CMV specific T-cell product is then infused back to the patient to elicit antiviral efficacy. One primary advantage to this approach is risk minimization for graft-versus-host disease (GVHD) or graft rejection (GR) as this product is matched to the recipient marrow, however comes at the expense of cost and VST generation time.

*Allogeneic, ex vivo expanded VST (aka. Off-the-Shelf VST)*: in this approach, a pool of VST products from varying HLA haplotypes is generated. The methodology for VST generation in this approach bears similarity to autologously expanded VSTs in that collected PBMCs undergoes repeated antigenic and cytokine stimulation to generate an expanded VST product ex vivo [15,16,32,33]. The VST product is then cryopreserved for future infusion upon demand. Several advantages to this approach are noted. While the generation of each cell product does require time, this is done in advance; cryopreservation of a VST with known HLA and anti-viral activity allows for rapid product identification and infusion. This approach also allows for development of multi-virus (i.e., combination of CMV, EBV, Adenovirus, HHV-6, BK, etc.) specific VST products for treatment of patients with polyviral detection [15,16,33]. Lastly, VST efficacy may be achieved with as little as 1 HLA match [15], improving the potential donor pool. Downsides to this approach also exist, primarily as some VST candidates may not have an available HLA-matched product among the banked VST pool. Cost also remains a large barrier to development as a large number of VST products will need to be generated to cover a broad portion of HLAs among the population that may require therapy.

*Donor derived, antigen selected VST*: in this approach, the original marrow donor, a matched donor, or partially matched donor, undergoes evaluation for the presence of prior viral infection (i.e., presence of CMV IgG antibody) for the virus of interest. The selected donor undergoes leukapheresis, for which PBMCs are collected and subsequently exposed to antigenic stimulation (ex. pp65 in setting of CMV) for IFNɣ production. Viral-specific T-cells are then *selected* utilizing a high throughput, IFNɣ capture system [18,34,35]. The resulting VST product is assessed for release criteria and infused into the patient. Several advantages exist with this approach, namely the relative rapidity of VST administration (typically about 1–2 weeks based upon donor screening and selection, with only an ~24-h time for VST generation from the donor leukapheresis). Improved HLA matching may also be achieved through this process than via use of an off-the-shelf product (i.e., utilizing parent or children as potential donors, thus achieving HLA haploidentical matching). Some downsides to this approach exist–it requires the potential donor to have prior viral T-cell immunity to the pathogen of interest (prior viral infection to CMV, for example) as the VST is a selected product that does not undergo antigenic exposure to *create* T-cell immunity. Similarly, as the VST product is not expanded, the total VST dose is often lower with this approach. Cost remains a relative barrier, as it requires investment in facilities and equipment for VST generation.

## 4. Adoptive Immunotherapy in HCT Recipients

The use of VSTs for treatment of CMV infection in HCT recipients has been well documented, ranging from case reports to phase I and II trials. Studies highlighting the use of CMV VSTs in HCT recipients are summarized in Table 1. In total, 26 studies and case reports, including >450 pediatric and adult HCT recipients with CMV infection, have been published detailing CMV VSTs as therapy. One of the first reports utilizing VSTs as therapy was by Einsele et al. [36], where 8 adult HCT recipients with refractory or resistant CMV infection received donor-derived CMV VSTs generated by ex vivo expansion. Enrolled patients received a VST cell dose of 1 × 10^7^/m^2^, and in total 6 of 8 (75%) patients demonstrated virologic clearance. Importantly, no evidence of de novo GVHD was reported in these patients. Cumulatively, clinical response rates of VSTs for treatment of CMV infection or disease ranged from 33–100%. However, in considering studies reporting ≥5 patients (21 in total), 86% (18/21) documented response rates of >70% (inclusive of partial or complete response to therapy). When considering efficacy in this context, indication for treatment is quite relevant-the primary indication of CMV VSTs in the majority of studies was refractory or resistant CMV infection, thus representing individuals who already failed standard-of-care antiviral therapy with ganciclovir, valganciclovir, or foscarnet. Numerous open trials for HCT recipients are available (clinicaltrials.gov, accessed on 4 October 2022) with varying target populations, modalities of VST generation, dosing regimens, and outcome measures. Several ongoing clinical trials are underway utilizing multi-virus specific off-the-shelf VST (NCT04013802) products or cytokine capture VST products (NCT04364178) for management of CMV infection after HCT. Importantly, one phase III randomized, controlled trial (NCT04832607) addressing VST therapy for CMV infection after HCT, is underway.

Use of VSTs as antiviral prophylaxis after HCT has also been explored. The first study to report adoptive transfer of CMV specific immunity for prevention was Riddell et al. in 1992 [37], utilizing donor-derived, MHC class I restricted CD8+ T-cell clones expanded ex vivo prior to infusion. Among the 3 patients who received therapy, none developed CMV reactivation post-transplant, which was accompanied by a measurable CMV specific T-cell response post-transplant. In total, 11 reports describing CMV VSTs as prophylaxis against CMV infection have been reported in the literature, inclusive of >100 pediatric and adult HCT recipients. Interpretation of cumulative efficacy in the prophylaxis data challenging based upon various factors, including risk of CMV reactivation, timing of VST administration (relative to detection of CMV) and outcome measures. In total, a wide range of efficacy with respect to viral reactivation was reported (0–100%). However, among 5 studies reporting rates of CMV reactivation post CMV VST administration [17,38,39,40,41], CMV detection after VSTs as prophylaxis occurred in ~36% (23/64) of patients. In addition to CMV detection after VST administration, other endpoints have been assessed. Blyth et al. [17] reported on VST administration for prevention of CMV infection or reactivation after allogeneic HCT in adult and pediatric patients. In their study, donor-derived, ex vivo expanded CMV VSTs were administered on D+28 at doses of 2 × 10^7^/m^2^. While their study found no difference in the rate of reactivation between individuals who received VSTs and control patients (*p* = 0.17), the study did note a significant reduction in peak viral load (*p* = 0.04) and number of patients who required antiviral therapy (36% versus 18%, *p* = 0.01) [17].

**Table 1 viruses-14-02370-t001:** Adoptive Immunotherapy for Treatment or Prophylaxis of Cytomegalovirus Infection or Disease in HCT Recipients.

Author	Year	Patient Age	Number of pts	Population	VST Source	Indication	Cell Dose (Total, /m^2^, or /kg)	Results	Adverse Events
Pei et al. [42]	2022	Adult, peds	190	Haplo HCT	Donor	Refractory CMV or at risk for progressive CMV disease	1 × 10^3^/kg–1 × 10^9^/m^2^	Cumulative response at 6 weeks: 89.5%	No infusional toxicities, cumulative incidence of GVHD at 3 months 14.3%
Rubinstein et al. [43]	2022	Peds	23 (CMV; *n* = 13 with either D+ or R+ serostatus)	Allo HCT	Donor	Prophylaxis of infection (CMV, EBV, AdV, BK)	2 × 10^7^/m^2^	3 patients with clinically significant viremia (CMV = 1)	2 patients with GVHD
Wang et al. [44]	2021	Peds	10	Allo HCT	Donor	CMV disease or refractory infection	0.5–1 × 10^8^/kg	All patients with improved viral load at 4 weeks	No exacerbation of existing GVHD
Gottleib et al. [45]	2021	Adult	11	Allo HCT	Donor	Prophylaxis of infection (CMV, EBV, AdV, VZV, Influenza, BK, A. fumigatus)	Not mentioned	8 of 10 patients with CMV reactivation noted before infusion, 3 developed end-organ disease	No infusional toxicities, 6 cases of GVHD post infusion
Celilova et al. [46]	2020	Peds	1	Allo HCT	3rd party	Drug resistant CMV infection	2 × 10^4^/kg	Initial virologic clearance, though recrudescent disease observed	No infusional toxicity, no GVHD
Seo et al. [47]	2019	Peds	1	Allo HCT	3rd party (Haplo)	CMV infection or disease	2 × 10^7^/m^2^	Clinical and virologic improvement	No infusional toxicity, no GVHD
Abraham et al. [41]	2019	Peds	14	Allo HCT	Donor	CMV disease or CMV prophylaxis	5 × 10^6^–2.5 × 10^7^/m^2^	3/4 with CMV viremia with virologic improvement, 6/7 with prophylaxis without reactivation	GVHD in 6 patients in study
Alonso et al. [48]	2019	Peds	2	Allo HCT	3rd party (Haplo)	CMV infection or disease	0.8–4.4 × 10^4^/kg	Clinical and virologic improvement	No infusional toxicity, no GVHD
Kallay et al. [49]	2018	Peds	9 (CMV = 3)	Allo HCT	3rd party	CMV infection or disease	7.5–16.2 × 10^4^/kg	2/3 (66%) with complete or partial virologic resolution	No GVHD, no infusional toxicities.
Withers et al. [50]	2017	Adult, peds	30	Allo HCT	3rd party	CMV infection or disease	2 × 10^7^/m^2^	28/30 (93%) with complete (23) or partial (5) response at ~8 weeks	GVHD (grade II) in 2 patients
Neuenhahn et al. [51]	2017	Not reported	16	Allo HCT	Donor, 3rd party	Refractory CMV infection or disease	6.6 × 10^6^ to 1.4 × 10^7^	12/15 (80%) patients with response	Fever in 1 patient post infusion, GVHD (grade II-III) in 2 patients
Tzannou et al. [16]	2017	Adult, peds	54 (CMV = 17)	Allo HCT	3rd party	Infection due to CMV, EBV, AdV, BK, or HHV-6	2 × 10^7^/m^2^	Complete and partial response rate: 94.1% by week 6	Fever with 1 patient, otherwise no infusional toxicities. 6 (11%) patients with new or recurrent GVHD post infusion
Pei et al. [52]	2017	Adult, peds	32	Haplo HCT	Donor	Refractory CMV infection	Median ~2 × 10^7^	27/32 (84%)	1 patient with grade II GVHD (had grade I GVHD prior to infusion)
Naik et. al [53]	2016	Adult, peds	36 (CMV = 7)	Primary immune deficiency, s/p HCT (35/36)	Donor, cord blood, or 3rd party	Infection with CMV, EBV, AdV, BK, or HHV-6	5 × 10^6^–1.35 × 10^8^/m^2^	Complete or partial response in 86% of pts with CMV (*n* = 7)	5 total events, GvHD (*n* = 4) associated with weaning IST.
Creidy et al. [54]	2016	Adult, peds	15 (CMV = 10)	Allo HCT	Donor	CMV disease, AdV disease	Median 3.5 × 10^3^/kg	3 of 9 evaluable patients with decreased viral load.	GvHD in 1 patient
Koehne et al. [55]	2015	Adult, peds	16	Allo HCT	Donor	Refractory CMV infection	0.5–2 × 10^6^/kg	14/16 patients with viral load clearance	No infusion related toxicity. No De novo GvHD.
Ma et al. [40]	2015	Adult	10	Allo HCT	Donor	Prophylaxis for CMV, VZV, AdV reactivation or infection	2 × 10^7^/m^2^	6 patients developed reactivation, however only 1 required therapy	2 cases of De Novo GVHD post T-cell infusion
Clancy et al. [56]	2013	Adult	7	Allo HCT	Donor	Prophylaxis for CMV reactivation or infection	2 × 10^7^/m^2^ at 28d post HCT	Unable to assess (3 patients with CMV detection prior to infusion)	No infusion related toxicity, GVHD in 2 patients post infusion
Leen et al. [15]	2013	Adult, peds	50 (CMV = 23, 19 evaluable)	Allo HCT	3rd party	Infection of AdV, CMV, EBV	2 × 10^7^/m^2^	CMV: response rate at 6 weeks: 74%	All infusions well tolerated. De novo GVHD (n = 2)
Gerdemann et al. [32]	2013	Adult, peds	10 (CMV = 5)	Allo HCT	3rd party	Infection of AdV, CMV, EBV	0.5–2 × 10^7^/m^2^	~80% success rate with respect to CMV	No infusion related events. GVHD in 1 patient.
Blyth et al. [17]	2013	Adult, peds	50	Allo HCT	Donor	Prophylaxis for CMV reactivation or infection	2 × 10^7^/m^2^ after 28d post HCT	Reduction in need for directed CMV therapy (GCV, foscarnet), no reduction in reactivation rates	aGVHD in 12/50 (5 prior to VST infusion).
Bao et al. [57]	2012	Adult, peds	7	Allo HCT	Recipient	Refractory CMV viremia	2.5–5 × 10^5^/kg	6/7 with partial or complete virologic response	No GVHD
Meij et al. [58]	2012	Adult, peds	6	Allo HCT	Donor	Refractory CMV viremia	0.9 × 10^4^–3.1 × 10^5^/kg	Complete virologic response in all patients	No infusion related toxicities, no GVHD
Schmitt et al. [59]	2011	Adult	2	Allo HCT	Donor	CMV viremia	0.37 × 10^5^/kg–2.2 × 10^5^/kg	Virologic resolution	No infusional related toxicities, no GVHD
Peggs et al. [35]	2011	Not stated	18	Allo HCT	Donor	Preemptive or prophylaxis of CMV	1 × 10^4^/kg	7 patients with prophylaxis administration–no antiviral therapy required. No significant CMV related disease observed.	No infusion related toxicity, GVHD in 3 patients
Feuchtinger et al. [18]	2010	Adult, peds	18	Allo HCT	Donor	Refractory CMV reactivation or disease	≤5 × 10^4^/kg (Mean 2.1 × 10^4^)	15/18 cases with clearance of viremia or significant reduction (>1 log)	No infusion related events. GI bleed in 1 patient
Micklethwaite et al. [38]	2008	Adult	12	Allo HCT	Donor	Prophylaxis for CMV infection	2 × 10^7^/m^2^ on D28 post HCT	4 patients with CMV post infusion, though overall low level	4 patients with GVHD post infusion (most associated with subtherapeutic IST levels)
Horn et al. [60]	2008	Adult	1	Haplo HCT	Donor	Refractory CMV infection	2.5 × 10^5^	Virologic improvement	No GVHD or infusion related events
Mackinnon et al. [61]	2007	Not stated	23 (16 with expanded product, 7 with IFNɣ selected product)	Allo HCT	Donor	CMV reactivation or disease	1 × 10^5^/kg (expanded) or 1 × 10^4^/kg (IFNγ selected)	8 (50%) with virologic resolution (expanded), 4/5 (80%) with reduced antiviral duration of therapy (IFNɣ selected)	No infusional toxicities, 3 patients with grade I GVHD
Leen et al. [33]	2006	Adult, peds	11	Allo HCT	3rd party	CMV reactivation or disease	5 × 10^6^–1 × 10^8^/m^2^	All subjects with prolonged CMV antigen clearance	No GVHD observed in cohort
Perruccio et al. [39]	2005	Adult, peds	25	Haplo HCT	Donor	Prophylaxis of CMV infection	1–10 × 10^5^/kg	7/25 with CMV reactivation, 2 deaths	GVHD (grade II) in 1 patient
Cobbold [62]	2005	Not stated	9	Allo HCT	Donor	Refractory CMV viremia or primary detection	1.2 × 10^3^/kg–3.3 × 10^4^/kg	Viremia resolution in 8/9, viral load reduction in 1.	GVHD (grade II) in 2 patients (though grade I GVHD noted prior to infusion)
Peggs et al. [63]	2003	Not stated	16	Allo HCT	Donor	CMV infection	1 × 10^5^/kg	All 16 cleared viral detection (8 while also receiving antiviral therapy). 2 episodes of CMV recurrence noted	No infusion related toxicity. GVHD in 3 patients (De novo in 2)
Einsele et al. [36]	2002	Adult	8	Allo HCT	Donor	Refractory CMV infection	1 × 10^7^/m^2^	6/8 with virologic clearance	No GVHD events
Walter et al. [64]	1995	Adult, peds	14	Allo HCT	Donor	Prevention of CMV infection	3.3 × 10^6^–1 × 10^9^/m^2^	No patient developed CMV viremia or disease	GVHD (grades I-II) in 3 patients
Riddell et al. [37]	1992	Adult	3	Allo HCT	Donor	Prevention of CMV infection	3.3 × 10^6^–1 × 10^9^/m^2^	No patient developed CMV viremia or disease	No infusional toxicity observed

Abbreviations: HCT, hematopoietic cell transplantation; VST, viral specific T-cell; Haplo, haploidentical; CMV, cytomegalovirus; GVHD, graft-versus-host disease; Allo, allogeneic; EBV, Epstein–Barr virus; AdV, Adenovirus; VZV, varicella zoster virus; HHV-6, human herpesvirus 6; aGVHD, acute graft-versus-host disease; IFNγ, interferon gamma.

Currently, one randomized clinical trial (NCT02108522) is underway studying the effect of an off-the-shelf, multi-virus specific T-cell product (ALVR105) for prevention of viral infection, including CMV, after allogeneic HCT. The VST product in this study was utilized for treatment of CMV infection post HCT in a Phase II, non-randomized open-label study with noted treatment success [16].

With respect to potential adverse events, the greatest concern was the development of GVHD after VST infusion. Reported rates of GVHD ranged considerably, from 0–22% among patients receiving the VST as therapy, in contrast to 4–55% who received the VST as an antiviral prophylaxis. In evaluating cumulative rates of GVHD from trials, individuals receiving VSTs as prophylaxis tended to have higher rates of GVHD post-VST infusion in comparison to those with receipt as treatment (~25% versus ~10%); however, this finding is likely confounded by the timing of VST administration post-transplant, and difficulty in differentiating GVHD development related to the VST product as opposed to that of HCT. Importantly, the grade of GVHD post-VST infusion was often mild in nature (grade I-II), with very few reports of severe morbidity or mortality post-VST infusion.

## 5. Experience of Adoptive Immunotherapy in SOT Recipients

The application of adoptive immunotherapy among SOT recipients is less robust than HCT recipients. The majority of patients treated with this approach have utilized Epstein–Barr Virus (EBV) specific VSTs in treatment of EBV+ PTLD disorders or viremia with variable success [65,66,67,68,69]. Studies highlighting use of CMV VSTs in SOT recipients are summarized in Table 2. The majority of reports are single cases describing success, however a recent phase I trial among adult SOT recipients noted success of autologous, ex vivo expanded CMV VSTs in treatment of CMV infection. [31] In this study by Smith et al., 13 adult SOT (lung, heart, and kidney) recipients received CMV VSTs; 11 (84.6%) demonstrated clinical or virologic response to therapy. Importantly, administration of autologously derived CMV VSTs were able to demonstrate efficacy despite infusion in the setting of ongoing maintenance immunosuppression, as well as avoiding the development of graft rejection or dysfunction. Given the potential of delivering alloreactive T-cells in SOT recipients, graft rejection is among one of the greater concerns with VST therapy. Fortunately, only 1 case report of fatal graft rejection post VST administration for CMV infection has been noted [70], however the total number of SOT recipients who’ve received CMV specific VSTs remains sparce in the literature.

## 6. Experience of Adoptive Immunotherapy in Other Immunocompromised Patient Popu-Lations

Limited data exist for use of adoptive immunotherapy beyond the HCT and SOT recipient. Patients with primary immunodeficiencies have received VSTs for treatment of CMV infection prior to subsequently after transplantation with complete or partial response in 86% of cases [53]. Recently, the use of a haploidentical, selected VST productfor treatment of CMV disease in a patient with rheumatoid arthritis and chronic lymphocytic leukemia was reported, noting clinical improvement after therapy [74].

## 7. Challenges, Limitations, and Future Directions of Adoptive Immunotherapy

Several limitations exist regarding use of adoptive immunotherapy. Due to the cellular nature of the VST product, medications with effects against T-cell activity or number have direct impact on the product. Several of these effects have been directly accounted for in the study design of protocols, specifically limiting dosing of corticosteroids to ≤0.5 mg/kg/d of prednisone equivalent, and the temporal administration of T-cell depleting agents (such as anti-thymocyte globulin and alemtuzumab, generally within 21–28 days of T-cell infusion). Ongoing immunosuppression, such as tacrolimus in HCT and SOT recipients, may also impact the overall efficacy of T-cell products. To address this, development of calcineurin-resistant VSTs have begun though have yet to enter clinical trials [75,76,77].

Several unanswered questions remain regarding VST therapies. One important question revolves around the durability of the infused product within the host. Leen at al. studied this by evaluating T-cell receptor rearrangement to identify the infusional product among the host T-cell population, noting persistence of the infusional product for up to 6 weeks in studied patients [15]. However, the duration of VST persistence varied among patients in this study. Amazingly, sequencing of T-cell receptor Beta chain (TCRB) to track VST persistence has noted detectable VST populations in recipients for up to 4 years post infusion [78]. Nevertheless, the duration of VST persistence remains relatively unknown, and is likely further impacted based upon qualities of the infusional product (ex. autologous versus HLA mismatched), as well as numerous recipient related factors (ex. presence of and treatment for GVHD/graft rejection, baseline immunosuppressive medications, etc.).

Several other critical questions remain regarding VSTs and the host, specifically “when” (i.e., timing) these cells should be administered and “who” is the appropriate candidate. The majority of clinical trials have focused on individuals with refractory or resistant CMV infection, providing therapy to those failing first line therapies before proceeding to VSTs. Contemporary studies have sought to include treatment for individuals earlier into the disease course, utilizing the VST option earlier alongside standard-of-care antiviral therapies. Several studies have included the use of VSTs as a prophylactic approach among HCT recipients. However, inclusion and exclusion criteria have varied substantially among studies, leading to difficulties in interpretating the appropriate target populations for VSTs. Additionally, variations in cell dose, use of multiple versus single-dosing approaches, use of multi-virus specific VST products, and other confounding factors (including administration of standard-of-care antiviral therapy, concomitant immunosuppressive therapies, etc.) make interpretation of efficacy challenging. Lastly, and importantly, publication bias may impact the appearance of VST efficacy.

To address this, randomized, placebo-controlled trials are needed to evaluate the true efficacy and safety of VSTs. One such trial is underway for CMV infection after allogeneic HCT (NCT04832607), and several others are enrolling for alternative disease processes such as BK polyomavirus (NCT04390113) and Adenovirus (NCT05179057). Lastly, an additional trial assessing prophylaxis against CMV, EBV, Adenovirus, BK and JC polyomaviruses, and human herpesvirus 6 (NCT05305040) in currently enrolling. Importantly, these last three mentioned clinical trials are all utilizing the same partially HLA-matched, off-the-shelf multi-virus VST product (ALVR105) for study. Additional randomized, controlled trials evaluating the various VST generation methodologies (ex. autologously derived VST, antigen-selected VST, multi-virus versus single-virus specific) in context of the degree of HLA-matching, dosing regimens (ex. total dose, single versus multiple dosing), and recipient characteristics (ex. HCT versus SOT) are needed to assess true efficacy and safety among the varying clinical contexts. Nevertheless, despite these limitations, adoptive immunotherapy has shown promise in the setting of prophylactic administration as well as for treatment of CMV infection, particularly in settings of refractory or resistant CMV disease.

## Figures and Tables

**Table 2 viruses-14-02370-t002:** Adoptive Immunotherapy for Treatment of Cytomegalovirus Infection or Disease in SOT Recipients.

Author	Year	Patient Age	Number of pts	Population	VST Source	Indication	Cell Dose (Total, /m^2^, or /kg)	Results	Adverse Events
Miele et al. [71]	2021	Peds	1	SOT (liver)	3rd party donor)	CMV disease (pneumonitis)	1 × 10^6^/kg	Decreased viral load,	No infusion related toxicity, graft function stable
Smith et al. [31]	2019	Adult	21 enrolled (13 received therapy)	SOT Renal, 4 Lung 8 Heart 1	Recipient	CMV disease or persistent viremia	1–2 × 10^7^/m^2^	11/13 (84.6%) with virologic or clinical improvement.	No infusion related toxicities, graft function stable
Macesic et al. [72]	2015	Adult	1	SOT (renal)	3rd party donor	CMV disease	1.6 × 10^7^/m^2^	Decreased viral load, asymptomatic	No infusion related toxicity, graft function stable.
Holmes-Liew et al. [73]	2015	Adult	1	SOT (lung)	Recipient	CMV disease (hepatitis)	12 × 10^7^ divided over 4 doses	CMV viral load undetectable, resolved end organ disease	No infusion related toxicity, no graft rejection
Brestrich et al. [70]	2008	Adult	1	SOT (lung), CVID pre SOT	Recipient	CMV disease (pneumonitis)	1 × 10^7^/m^2^	Symptomatic improvement, decreased viral load	No infusion related toxicity, later death from graft failure

Abbreviations: SOT, Solid organ transplantation; VST, viral specific T-cell; CMV, cytomegalovirus; CVID; common variable immune deficiency.

## Data Availability

Not applicable.

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
