# Peer review of "Adoptive Immunotherapy for Prophylaxis and Treatment of Cytomegalovirus Infection"

_viruses, 2022, doi:10.3390/v14112370_

Round 1
Reviewer 1 Report
Adoptive immunotherapy for CMV and other infections has huge clinical potential. The review article “Adoptive Immunotherapy for Prophylaxis and Treatment of Cytomegalovirus infection” provides a clear and systematic overview of the clinical use of cytomegalovirus-specific T-cell therapy for infections during hematopoietic cell transplantation and solid organ transplantation. Overall, the manuscript is concise, well-written, and well-structured, relevant literature was included and can be accepted in this form.
Author Response
I appreciate the comments from the reviewer. I have made some additional modifications to the manuscript to address grammatical errors and improve the quality of the manuscript.
Reviewer 2 Report
In this report, Christopher P. Ouellette discusses advancements, some encouraging results and challenges of widespread adoption of viral specific T cells in the prophylaxis of CMV infection of immunocompromised patients. The review is comprehensive and sufficiently critical to be useful for researchers and practitioners working in the field. A major drawback is the lack of graphical presentation which may be very helpful for those that are not strictly involved in the specific part of the field. I would also suggest including VST abbreviation explanation in line 95 because the explanation is present only in the abstract.
Author Response
I appreciate the comment from the reviewer. I have addressed the question of VST abbreviation by making notation earlier in the manuscript (line 63 in edited version). I do agree with the reviewer re: lack of a visual representation - I admittedly struggled to think of a visual representation that would be beneficial to those not in the field, especially ones not used in other manuscripts addressing this topic (referenced in this study). I would gladly accept suggestions of beneficial visual representations and include a graphic as able, especially if felt of benefit to the journal's readership.